# Biodeterioration of Microplastics: A Promising Step towards Plastics Waste Management

**DOI:** 10.3390/polym14112275

**Published:** 2022-06-02

**Authors:** Aatikah Tareen, Saira Saeed, Atia Iqbal, Rida Batool, Nazia Jamil

**Affiliations:** 1Department of Microbiology and Molecular Genetics, The Women University Multan, Multan 66000, Pakistan; atikatarin@gmail.com (A.T.); sairasaeed16@yahoo.com (S.S.); atia.iqbal@wum.edu.pk (A.I.); 2Institute of Microbiology and Molecular Genetics, University of the Punjab, Lahore 54590, Pakistan; rida.mmg@pu.edu.pk

**Keywords:** micro-pollutants, landfill, soil biota, polyethylene, polyester

## Abstract

Polyethylene and Polyester materials are resistant to degradation and a significant source of microplastics pollution, which is an emerging concern. In the present study, the potential of a dumped site bacterial community was evaluated. After primary screening, it was observed that 68.5% were linear low-density polyethylene, 33.3% were high-density, and 12.9% were Polyester degraders. Five strains were chosen for secondary screening, in which they were monitored by FTIR, SEM and weight loss degradation trials. Major results were observed for *Alcaligenes faecalis* (MK517568) and *Bacillus cereus* (MK517567), as they showed the highest degradation activity. *Alcaligenes faecalis* (MK517568) degrades LLDPE by 3.5%, HDPE by 5.8% and Polyester by 17.3%. *Bacillus cereus* (MK517567) is better tolerated at 30 °C and degrades Polyester by 29%. Changes in infrared spectra indicated degradation pathways of different strains depending on the types of plastics targeted. Through SEM analysis, groves, piths and holes were observed on the surface. These findings suggest that soil bacteria develop an effective mechanism for degradation of microplastics and beads that enables them to utilize plastics as a source of energy without the need for pre-treatments, which highlights the importance of these soil bacteria for the future of effective plastic waste management in a soil environment.

## 1. Introduction

After their development over the last hundred years, plastics have been utilized in numerous ways, and are associated with every part of our lives [1]. Regardless of their benefits, issues identified with their wide application cannot be ignored. Most of these plastics are resistant and remain in the earth for generations [2]. According to recent information, worldwide plastic production reached 360 million tons in 2018. Asia is the biggest maker and consumer of the world’s plastics. Among worldwide plastic producers, China is the largest (30%), trailed by Europe (17%), and in total 18% are from Canada, Mexico and US [3]. It is appropriate to specify that petrochemical plastics account for over 80% of overall plastic use. For instance, polyethylene terephthalate, polyester, polyethylene, polypropylene, polystyrene, and polyvinyl chloride are the most commonly utilized plastics [4]. Various strategies utilized for plastic waste management are dumping in a garbage lot, burning and reusing. Each of these techniques has its disadvantages; for example, plastics when burnt or incorporated into landfills utilize sources of land, which causes pollution, and limited natural resources are wasted. Currently, limited quantities are reused and this causes a hindrance to the idea of the circular economy; only 9% of plastic has been recycled from the total production of 8.3 bn metric tons since the 1950s [5,6]. It has also been reported that in 1990 Coca-Cola promised to use 25% recycled plastic by 2015, but in 2020 only 11.5% had been recycled [7].

Among the plastics which currently exist, fragmented particles have gained attention and are even more troubling. These are characterized as microplastics (1 μm–5 mm) based on their diverse range [8,9]. Due to the smaller size and wide dispersion of these fragments, it is hard to gather them and remove them subsequently from the environment. In the same way, the speed at which they enter nature outstrips their clearance speed [10]. Besides, microplastics are found in potable water, whichmay become a danger to wellbeing [11]. Polyethylene is a generally utilized type of plastic because of its conveyance ability. Nonetheless, because of its saturated linear hydrocarbon chains, and because it can be communicated as -[CH2-CH2], PE items are hard to be corrupted by nature [12]. A feasible way-out of this problem is conversion of plastics by microbes. Microbes that have the potential to degrade plastics belong mostly to the phyla Proteobacteria, Firmicutes and Actinobacteria, and most were screened from polluted dumpsites. Studies should concentrate on biodegradation of microplastics focusing on the most well-known contaminants such as polyethylene [13].

Microbial assisted deterioration of polymers by bacteria and fungi has been effectively researched in indigenous habitats, such as soil. For example, *Bacillus* sp., *Rhodococcus* sp. [14], *Pseudomonas aeruginosa* [15], *Zalerionmaritimum* [16], and *Aspergillus clavatus* [17] can utilize plastic polymers as their energy source in cultivation media, therefore causing a decrease in the dry weight of polymers and inciting physical–chemical alterations which include modifications in surface morphologies. Subsequently these microbes form cracks and rough surfaces and produce chemical bonding structures, such as carbonyl groups, ketones and aldehydes [18]. Microorganisms can break down the polymers in two phases, firstly inside them, and then becoming enzymatically dissimilated in order to discharge extracellular proteins, causing chain cleavage in the monomer, which can be used by microorganisms [14]. Microbial species have the ability to change polymers into monomers and, additionally, to dissimilate into carbon dioxide and water. The biodegradation accomplished by microbes is related key characteristic such as the atomic weight and crystallinity of the polymers [19,20,21].

The aim of the present study is to evaluate the potential of locally isolated strains to degrade plastics with different chemical compositions based on their petrochemical origin. *Alcaligenes faecalis*, *Bacillus* spp. and *Staphylococcus* sp. were isolated from dumps in Multan, with affinity for LLDPE, HDPE and Polyester and with a considerately faster decay rate which is globally needed, especially in developing countries.

## 2. Material and Methods

### 2.1. Polyethylene and Polyester Used in the Current Study

Plastic beads and powders of Linear low-density polyethylene, high-density polyethylene and aromatic Polyester used in the current study were provided by Mehran Plastic Industries Pvt Limited, Karachi, Pakistan. LLDPE, HDPE and polyester (HOROH) and terephthalic acid (p–HOOC–C_6_H_4_COOH) were in the form of fine powders with 99.5% purity. Beads were sterilized, followed by washing with 70% ethanol and drying with filter paper to make them ready-to-use in experimentation.

In this study, biodegradation of microplastics and beads was carried out by soil biota. Figure 1 graphically explains the research framework.

### 2.2. Isolation, Identification, and Screening of Bacterial Isolates

Samples of soil (5 g) were collected from different municipal dumpsites in Multan which are heavily contaminated with various plastic waste, in commercial areas. These specified areas were chosen since they had been used as a plastic dump site for a very long time, expanding the likelihood of finding bacteria that has the ability to degrade plastics. By using a conventional serial dilution method, 50 µL from 10^−6^ dilution was spread on a nutrient agar (Sigma Aldrich, Saint Louis, MO, USA) plate infused with 1% of plastic powder for the isolation of plastic degrading bacteria, and for each type of given plastic this procedure was repeated. After 15 days of incubation at 37 °C, bacteria with distinct zones were selected and sub-cultured for further characterization [10]. Based on morphological characterization of the bacterial colony after primary screening, Gram-stained slides were observed under light microscope at 40× and 100× for identification.

Secondary screening of isolates was performed by both static and shaking methods using mineral salt medium previously used by Osman et al. [22] supplemented with 5% polyethylene (LLDPE and HDPE) and polyester powder. Isolation of plastic degrading bacteria was verified by streaking the bacteria on 5% concentration solid media, and diameter of zone was observed after 15 days of incubation at 37 °C, while shaking technique involved biodegradation assay and liquid media. After secondary screening, molecular identification was\performed by amplification of 16S rRNA gene by using primers 785F 5′ (GGA TTA GAT ACC CTG GTA) 3′, 27F 5′ (AGA GTT TGA TCM TGG CTC AG) 3 from the Macrogen sequencing system. Sequences were analyzed utilizing the BLAST tool from the National Centre for Biotechnology Information (NCBI) (https://www.ncbi.nlm.nih.gov/ accessed on 25 May 2022) against reference 16S rRNA sequences of type strains and submitted to GenBank. To construct consensus, neighbor joining tree phylogenetic analysis was carried out by using Molecular Evolutionary Genetics Analysis (MEGA) version 7 software (Pennsylvania State University, State College, PA, USA) [23].

### 2.3. Rate of Deterioration by Weight Loss Strategy and the Impact of Temperature and pH on Weight of Plastic

Isolates showing zone of clearance in static secondary screening were further assessed in the degradation study. Each isolate was separately inoculated in flasks containing 20 mL of minimal medium and beads of Polyethylene (LLDPE and HDPE) and Polyester, separately. The biodegradation study was implemented for a period of 40 days. Analysis was caried out by weight reduction using the following formula [24].
Weight loss % = initial weight − final weight/initial weight × 100(1)

Biodegradation study of LLDPE, HDPE and Polyester was performed at different temperature and pH conditions in order to optimize the isolated bacterial cultures in different environmental conditions, for future study. The range of temperature used was 30–45 °C, and the pH was maintained at 4.0 and 8.0 scale. The weight of plastic beads was recorded earlier. Optimization of isolates was carried out for 15 days [25].

### 2.4. Formulation of Microbial Consortia and Determination of Weight Loss for the Mixture of Plastic Pellets in Local Natural Conditions

An acclimated consortium capable of degrading polyester, high-density and linear low-density polyethylene pellets was developed in the laboratory. A 500 mL flask with nutrient broth (Sigma Aldrich, Saint Louis, MO, USA) inoculated with *Alcaligenes faecalis*, *Streptococcus* sp. and three types of *Bacillus* spp. was utilized for the degradation experiment. Subsequently, the consortium was cultured, until lag phase was developed, for 6 days. A control was set up containing just broth and beads of LLDPE, HDPE and polyester [26], to examine biodegradation of LLDPE, HDPE and Polyester in off-site conditions. The biodegradation of commercially available plastic beads under the attack of bacteria was studied using the soil burial method for an interval of three months. Pre-weighted beads were washed with 70% ethanol and added to soil. Sterilized pellets were added to soil along with nutrient broth and inoculum. Soil selected was clay loam with pH 8.0. The experiment proceeded from March to May, so temperature variations (23 °C to 45 °C) could be noted.

### 2.5. Analysis of Plastic Beads Degradation by Sturm Test

Plastic degrading bacteria can break down the long chains of polymers into monomers by their different activities, either by oxidation or enzymatic hydrolysis. CO_2_ evolved as a result of mineralization of plastics in aerobic conditions. Evolution of CO_2_ by bacterial isolates can be assessed by Sturm test applied in a modified way. Pre-weighted beads of LLDPE, HDPE and Polyester were added to a flask containing 5 mL of MSM broth, and then 1 mL of KOH (1M) was added. This was inoculated with 24 h old culture and incubated in a rotatory incubator for 25 days. Aerobic conditions were maintained in the rotatory incubator. After 25 days of incubation, the amount of carbon dioxidegenerated was calculated in the test and in the control. CO_2_ produced as a result of mineralization was trapped in a flask containing KOH. Barium chloride solution Bacl2 (1M) was added to this flask, and as a result barium carbonate precipitates formed. Precipitates were filtered on filter paper and dried at 50 °C in an oven for an hour. The weight of the precipitates indicates CO_2_ generation by test organism as their end product [27].

### 2.6. Surface Modification Analysis of Plastic Beads by Scanning Electron Microscopy

Degraded beads of aromatic polyester, and linear-low density and high-density polyethylene were set up on an aluminum-disk with a width of 1.2 cm using double sidedblack colored carbon tape. Sputtering of samples with a thin layer of gold in a vacuum chamber was performed using argon gas and an electric current of approximately 3 mA. Then labelled tests pellets were placed sequentially in the scanning electron microscope chamber and magnification was set to display images at 500×, 1000× and 10,000× with a TESCAN MERA 3 field emission scanning electron microscope [28].

### 2.7. Structural Changes in Plastic Beads—Analysisby Fourier Transform Infrared Spectroscopy

The LLDPE, HDPE and Polyester beads treated with isolated bacteria for 40 days were examined by FTIR. Background noise was eliminated by performing a blank scan in the frequency range of 4000 to 600 cm^−1^. Thus, the samples were scanned in the region of 400–4000 cm^−1^ at a resolution of 4 cm^−1^. The resultant spectrum included a plot of rate transmittance versus wave number, which was additionally investigated against the respective comparative controls [29].

## 3. Results

### 3.1. Isolation and Screening of Bacteria with Potential to Degrade Microplastics of LLDPE, HDPE and Polyester

In the present study, isolation of effective polyethylene (LLDPE, HDPE) and aromatic polyester with degrading bacteria from municipal landfill soil was carried out. The dump sites’ soil samples were collected from the historic Daulat gate of the city, Northern By-pass, Sher-shah Road and Shujabad Road. Morphologically distinct isolates were used to determine the ability to use plastic as carbon source. The experiments were conducted over a screening series. Primary screening led to the isolation of 54 bacterial isolates. Out of 54 isolates, 37 were considered as linear low-density polyethylene degraders (68.5%), 18 isolates were high-density polyethylene degraders (33.3%) and seven isolates were polyester microfiber degraders (12.9%) as shown in Figure 2. Out of 54 isolates, bacterial isolates which show distinct growth and zone were streaked on 5% concentration of each given type of microplastic and observed after the formation of clear zones. *Alcaligenes faecalis* (SA-5) shows a zone of clearance with 3 mm for polyester microfibers and 1 mm for LLDPE. *Bacillus cereus* (SA-68) shows a 6 mm diameter for LLDPE and a 2 mm for HDPE. *Bacillus* spp. (SB-14) shows a 4 mm diameter for LLDPE and a 0.25 mm for polyester microfibers. Another *Bacillus* spp. (SC-9) shows a 1.5 mm diameter for HDPE and a 3 mm for LLDPE microplastics. *Streptococcus* spp. (SC-56) shows a 5 mm zone diameter for LLDPE and a 0.5 mm diameter for HDPE microplastics. These five isolates show prominent zones and were tested further for biodegradation assay.

### 3.2. Assessment of the Polyethylene (LLDPE and HDPE) and Polyester Deteriorating Bacteria Based on Weight Loss Percentagein Ex-Situ and Laboratory Conditions

The degradation was determined by calculating the percentage of weight loss in polyethylene and polyester beads by isolated bacteria after 40 days of incubation. Linear low-density polyethylene degraded by SA-5 *(Alcaligenes faecalis*) (MK517568), SA-68 (*Bacillus cereus*) (MK517567), *Bacillus* sp. (SB-14 and SC-9) and *Streptococcus* spp. (SC-56) by 3.5%, 15%, 11.8%, 4.8% and 9.8%, respectively. High density polyethylene was degraded by SA-5 (*Alcaligenes faecalis*) (MK517568) *Bacillus* sp. (SB-14 and SC-9) and *Streptococcus* spp. (SC-56) by 5.8%, 11.7%, 3.8% and 13.7%, respectively. Polyester was degraded by SA-5 *Alcaligenes faecalis* (MK517568), *Bacillus* sp. (SB-14 and SC-9) by 17.3%, 9.4% and 5.8%, respectively. Difference between initial and final weight indicates the significant extent of polyethylene and polyester utilization by the bacteria, as shown in Figure 3. Replicates were used to gain statistical confidence as standard deviation was used, while control (no bacteria) showed zero percentage of degradation, and beads were floating in the media. In the treatment flasks, beads had settled down due to bacterial action.

After the biodegradation assay, bacteria were optimized. After 15 days of incubation with continuous shaking, maximum percentage reduction in weight or percentage loss in weight (% WL) of the polyester beads was recorded with *Bacillus cereus* (MK517567) (29.4 ± 0.05) at 30 °C, as the highest weight loss during the experiment. *Alcaligenes faecalis* (SA-5) revealed 6.6% polyester-degradation at 45 °C. Polyester degradation was carried out at 30 °C as well as at 45 °C by both isolates (SA-5 and SA-68), respectively. In addition, 3.5% high density polyethylene degradation was recorded by *Alcaligenes faecalis* (SA-5), and *Bacillus cereus* (SA-68) showed (6.6 ± 0.02) for HDPE. At 30 °C the highest weight loss was observed by *Bacillus cereus* (SA-68) for polyester, along with (8.0 ± 0.02) for LLDPE. *Bacillus* spp. (SB-14 and SC-9) and *Streptococcus* sp. did not showvery promising results in changing environmental conditions. SA-5 and SA-68 were considered as promising degraders and were identified by sequencing.

### 3.3. Determination of Weight Loss in Local Natural Conditions by Consortium

The percentage of weight loss in natural conditions was 4.3 ± 0.02, 4.8 ± 0.02 and 2.7 ± 0.02 for LLDPE, HDPE and polyester beads after 90 days of incubation, respectively. The weight loss percentage of plastics strips was 4.8 ± 0.02 and 4.9 ± 0.02 for LLDPE and HDPE after 90 days of incubation, respectively. At the end of incubation, plastic strips were easy targets for bacteria compared to beads, and the possible reason for this may be that bacterial strain utilized the released C compounds from plastic strips during the degradation. It is suggested that the result with this type of method was observed for the first time in this region, and will help in the degradation of plastic strips under different environmental conditions.

### 3.4. Characterization and Molecular Identification

The morphological characteristics of the bacterial isolates were identified by conventional methods. Gram staining of bacterial isolates show that they are Gram-positive rods and cocci. Growth on nutrient media with small, circular, flat, white colored, opaque or translucent colonies with smooth edges was observed on the plates. Biochemical testing revealed that isolates are catalase, indole and MRVP negative, while positive for starch hydrolysis. SB-14 gives a yellow butt and red slant with gas production. Based on the 16S rRNA gene sequencing and biochemical characterization, the bacteria isolated from the site mainly belonged to Bacilli from *Bacillus cereus* and *Alcaligenes*. As inferred by 16S rRNA gene analysis, strain SA-68, showed 99% similarity with corresponding gene sequences of reference strains *Bacillus cereus* SBMWI and strain SA-5 showed 100% similarity with *Alcaligenes* sp. (KX164437.1), respectively. Phylogenetic trees (Figure 4 and Figure 5) showed that the strain was clustered on separate branches, with reference strains belonging to respective genera. The GenBank nucleotide accession numbers were assigned to strain *Bacillus cereus* SA-68 is MK517567, respectively. Query sequence was designated as *Alcaligenes faecalis* with GenBank accession no. MK517568.

### 3.5. Degradation of Plastic Beads Confirmed by CO_2_ Production in Sturm Test

Carbon dioxide, evolved as a result of deterioration of polyethylene (LLDPE, HDPE) and polyester by *Alcaligenes faecalis* (SA-5), *Bacillus cereus* (SA-68), *Bacillus* spp. (SB-14 and SC-9) *Streptococcus* sp. (SC-56), was trapped and compared to the amount evolved in the case of biotic control under similar conditions. Evolved carbon dioxide calculated from media inoculated was 0.602 g, 0.427, 0.723, 0.524 and 0.205 g/5 mL CO_2_ g^−1^ of C, respectively, while the control flask indicated no precipitates.

### 3.6. Fourier-Transform Infrared Spectroscopy (FTIR) Analysis

The changes in spectral peaks due to biodegradation were determined using a FTIR (ATR-alpha Bruker) spectrophotometer. The degradation of linear-low density and high-density polyethylene and aromatic polyester was confirmed by the changes in spectra of the FTIR analysis. The pellets without any treatment served as control, and pellets of LLDPE, HDPE and Polyester treated with isolated *Alcaligenes faecalis*, *Bacillus* sp. and *Streptococcus* spp. in MSM broth acted as test. In the case of LLDPE, the results of this study demonstrated that LLDPE pellets showed four peaks in the range from 2900 to 715 cm^−1^, at 2913.92, 2846.83, 1462.50 and 718.05. The vibrational mood of peaks was observed in comparison with control, and when LLDPE pellets were treated with Alcaligenes faecalis (SA-5), they showed more vibrational shifts than with *Bacillus* spp. (SC-9). The first peak shifted to 2913.83 cm^−1^ which indicates the C–H stretching of the methyl group, the second peak shifted to 2846.50 cm^−1^ which indicates the C–H symmetric and asymmetric stretching of methylene (C–H_2_), the third peak shifted to 1462.05 cm^−1^ which indicates the C=C and replacement of carbonyl bond with amine bond, and the fourth peak shifted to 718.23 cm^−1^ which indicates the –C=C– stretching and the presence of alkene group as shown in Table 1. For HDPE, the spectral peaks were observed from 2900 to 715 cm^−1^ when treated with bacterial isolates. The four peaks were observed from control of HDPE at 2913.94 (C–H stretching–CH_3_), 2846.58(CH stretching–CH_2_), 1461.42(bending C–H bond of methylene) and 718.23 cm^−1^ (C–O). In case of treatments with bacterial isolates, the wave number of spectral peaks showed shifts up to 2914.37 cm^−1^ which indicates the vibrations in the stretching of C=C bond seen in alkanes. The second peak showed shifts to 2846.70 cm^−1^ which indicates the stretching of C–H bonds in methylene, and the absorbance range of 3000–2800 cm^−1^ corresponds to C–H stretching and the presence of alkanes. The third peak shifted to 1461.76 cm^−1^ which indicates the –CH_2_ stretching and presence of aromatics, and the band around it also corresponds to bending deformation. The fourth peak shifted to 717.82 cm^−1^ which indicates the rocking deformation of bonds. For Polyester, two treatments were given for SA-5 and SC-9. as both were showed biodegradation for polyester along with other treatments. The nine spectral peaks were observed in control of polyester at 2024.83 (C–H), 1709.16 (Carbonyl group), 1409.10 (C=C), 1339.65 (CH_3_), 1241.02 (Ar-O-R), 1097.75 (C–O), 1017.93 (C–O), 871.00 (Aromatic ring) and 720.01 cm^−1^ (Mono-substituted aromatic ring). With treatment, the peak shifted to 1710.84 cm^−1^ which indicates a change the carbonyl group’s polyester component and thus the cleavage of the ester bond. The other peak shifted to 1408.45 cm^−1^ which indicates the decrease in intensity of band leads to bond cleavage of C=C. The other band shifted to 1340.04 which indicates CH_3_ symmetrical bending. The fourth peak shifted to 1238.19 which indicates the asymmetrical bending of Ar–O–R. The fifth peak shifted to1088.58 which indicates the bond cleavage of C–O, and 1017.93 (control) shifted to 1016.01 which indicates the bond cleavage of C–O. The peak 871.00 (control) shifted to 870.14 which indicates the aromatic ring bend out of plane and the peak 720.01 (control) shifted to 719.4 which indicates the stretch of monosubstituted ring. The first peak, which disappeared after treatment with SA-5, indicates that cleavage of C–H bond occurred which shows the formation of new intermediate products, as shown in Figure 6.

### 3.7. Surface Modifications Confirmed by Scanning Electron Microscopic Analysis

In this study, properties were authenticated with SEM and FTIR analysis. Degradation and morphological changes in polyester, liner-low density and high-density polyethylene pellets after bacterial treatment were analyzed by scanning electron microscopy. As regards the length of all samples, degradation was observed due to the roughness of surfaces and formation of cracks/holes/scions. (Figure 7, Figure 8 and Figure 9). Bacterial cells attached to surfaces were also visualized on some tested pellets. Scanning electron micrographs revealed that isolates cause localized surface deterioration in the plastic pellets, while the outside of untreated plastics is flawless and smooth even following 40 days of incubation. LLDPE samples’ surface indicates bacterial attachment and bacterial actions on surface as shown at 500×, while HDPE and polyester are shown at 1000 and 10,000× for better understanding.

## 4. Discussion

Detrimental effects of plastic waste are increasing. Therefore, its elimination from the earth is fundamental. Among thermic, photooxidative, mechanochemical, and catalytic degradation strategies, biodegradation is considered as the best choice for plastic waste degradation on account of the minimal effort required and its eco-accommodating nature [30]. However, detailed characterization of proficient plastic-detiorating microorganisms and microbial compounds should be completed [31]. Different studies demonstrate the biodegradation of plastics by marine water isolates and rhizosphere samples, but this study isolated bacteria from municipal landfill sites, as these sites are capable of accommodating bacteria with high potential to degrade plastics and remain stable. The current study from Multan, Pakistan isolates 54 bacterial isolates with a potential to degrade different types of microplastics and beads, and to the best of the author’s knowledge no previous studies have been found om isolation of these bacteria from this region. Therefore, these indigenously isolated strains can be valuable input for combating plastic pollution waste. Earlier in 2013 from Islamabad, Pakistan, Shah reported polyester-polyurethane degrading bacteria [32]. Primary screening results on the growth of isolates along with zones on infused media with plastic powder, and repeated screening with 5% concentration, led to formation of clear zones ranging from 0.25 to 6 mm. Compared to this, in a study conducted from Iraq the zones ranged from 2.5–3.0 mm on 0.1% LDPE infused MSM, which implied that the bacterial colonies engendering a clear zone were capable of degrading polyethylene in minimal media [10]. In the treatment flasks, beads settled down due to bacterial action.Many studies report that film-type plastics required at least two months for biodegradation to occur [33], which is a very long time for testing and makes them unsuitable for identification of new bacterial strains. Alternatively, the surface area of beads is larger than that of plastic films, and the chance of bacterial attachment increases, which may speed up the reaction [34]. To test this, we prescribed 40 days of incubation for beads in liquid media, and both rate of biodegradation and efficiency were improved. The apparent degradation efficiency was assessed by weight loss of beads in MSM, *Bacillus cereus* degraded LLDPE by 15%. *Alcaligenes faecalis* degraded polyester by 17.3% and weight loss for other types of plastic was also noted as mentioned in the appropriate section. Weight reduction may have been due to the different metabolism rate of different species. Different studies have reported that the rate of degradation of polyethylene varied from 1.5 to 13% as according to different assays and types of microbes [35].Another study from India also supports that *Bacillus* spp. are splastic degraders by using *Bacillus amyloliquefaciens* as LDPE degraders [36]. All these studies indicate that these are degraders for specific types of plastic, but for this study *Alcaligenes faecalis* and *Bacillus cereus* showed the potential to degrade more than asingle type of plastic.Weight reduction was also optimized at different environmental conditions in the laboratory and outside, in which *Bacillus cereus* (SA-68) showed maximum tolerance and degraded polyester by 29% weight loss after 15 days of incubation. In another study, optimization of growth media was carried out in which pH of the media containing HDPE gradually decreased after 90 days of incubation from 7.3 to 5% [37]. We optimized bacteria for their growth and potential ability to degrade plastic in different environmental conditions, although different studies use *Alcaligenes faecalis*, *Bacillus* spp. and *Streptomyces* spp. as potential degraders. In another study, *Alcaligenes faecalis* was isolated from PET-treated soil and confirmed as plastic degrader by FTIR analysis [38]. Another study from Iraq used *Streptomyces* spp. as source for pollution control, but suggested it was best as an LDPE degrader isolated from soil [39]. More than these studies, the effectiveness of degradation noticed in this study was comparable to that of LDPE film degraded by mixed microbial cultures of *Bacillus* sp. and *Paenibacillus* sp., in which both bacteria together exhibit at 15% the highest degradation efficiency [36]. Studies have described that plastics can be degraded into monomers, and then these monomers are converted into CO_2_ and water. To test this, a Sturm test was performed with a modified procedure. In another study, the process used involved titration with thiosulphate and sodium hydroxide in the presence of barium chloride and KOH. The highest rate of evolution for LDPE was 6.28 g/L and the lowest was 1.19 g/L [27]. Compared to this, the present study involves 0.723/5 mL noted previously. Curiously, the surfaces of pellets were seen to have become uneven and filled with cracks and grooves1 [10]. After thorough observations in different studies, it has been demonstrated that microorganisms can change not only appearance but functional groups and other characteristics [40]. In this study, these properties were authenticated by SEM and FTIR analysis. Previous studies utilized SEM micrographs as an analytical tool to demonstrate erosions, cavities, and pores formed on plastic films in order to indicate the extent of colonization and degradation [41]. Curiously, authors describe the surfaces of pellets as een to have become uneven and filled with cracks and grooves [10]. Other studies also confirmed degradation by formation of biofilm on the plastic surface. In the study conducted in 2014 on LDPE degradation by *Bacillus amyloliquefaciens* isolated from municipal solid, SEM analysis revealed that both the strainsexhibited adherence and growth with LDPE, used as a sole carbon source [36]. Another study that used a photocatalytic technique for deterioration of polystyrene also indicates that cavities/holes increased in size after the treatment [42]. In the present study it was also observed that cavities or holes were produced in pellets. SEM analysis revealed surface erosion and bacterial adhesion by giving proof of the deterioration of the plastic beads because of the activity of the bacteria, and ensured the degradation capability of the bacteria. In addition to these analyses, isolated strains efficiently degrade beads and microplastics of various categories of plastic. Hence, the difference in biodegradation rate between LLDPE, HDPE and polyester may be due to the presence of specific enzymes or concentrations of different enzymes required for kinds of different plastic degradation. This demonstrates that, during growth of bacteria, there may be different metabolism rates and uptake of energy from plastic as a source of carbon [34]. PE microplastics can be categorized into HDPE and LDPE. Enzyme based degradation is divided into two steps, extracellular and intracellular. In the first step, LDPE is broken down into shorter chains and in the next is step followed by mineralization into CO_2_ and H_2_O as described in the Sturm test. Laccase and Alkane showed that the reaction in polyethylene belong to that of the AlkB family of enzymes, while Laccase is the most commonly reported enzyme responsible for HDPE degradation, and alkane hydrolase for LDPE degradation [13,43]. Other studies also confirmed that manganese peroxidase and laccase enzymes produced by *Bacillus cereus* were involved in the degradation of low-density polyethylene after an incubation of nine weeks followed by confirmation by FTIR [41]. Another study explained that *Alcaligenes faecalis* produced extracellular enzymes such as CMCase, protease, xylanase and lipase, which indicates that the surface of the plastic was attacked by these enzymes after SEM and FTIR analysis [44]. Montazer et al. explain that these secreted bacterial enzymes followed the formation of monomers by the β-oxidation system pathway [45].Spectral changes indicate the changes in formation of new bands at 1460, 600–700 cm^−1^. Our observations are similar to a previous report in which bands at 2900, 720, 1460 cm^−1^ indicated the rocking deformation and stretching of bonds [46]. Another study also explains the degradation of Polyester and observed FTIR peaks, and functional groups observed were similar to the present study as CO (2230–2050 cm^−1^), aliphatic compounds (3140–2640 cm^−1^), carbonyl compounds (1900–1680 cm^−1^) and alkene (940–850 cm^−1^) are identified [47]. In another study, it was observed through FTIR that addition of –OH group to LDPE backbone was due to the activity of bacteria producing enzymes for degradation of plastic [48]. In this study, observations were made which indicate that cleavage in the carbonyl group also suggests plastic degradation. In addition to these analyses, isolated strains efficiently degrade beads and microplastics of various categories of plastic.

## 5. Conclusions

The current investigation demonstrated the degradation of three types of plastic by soil bacteria isolated from municipal landfill sites. Among 54 isolates, *Alcaligenes faecalis* acted as most the promising candidate, but *Bacillus* spp. and *Streptococcus* sp. performed successful degradation, with stability. Bacterial attachment and formation of cracks were well observed via scanning electron microscope. The methods developed may help in future to screen and identify new bacterial strains that have the capability of degrading more than single type of plastic.

## Figures and Tables

**Figure 1 polymers-14-02275-f001:**
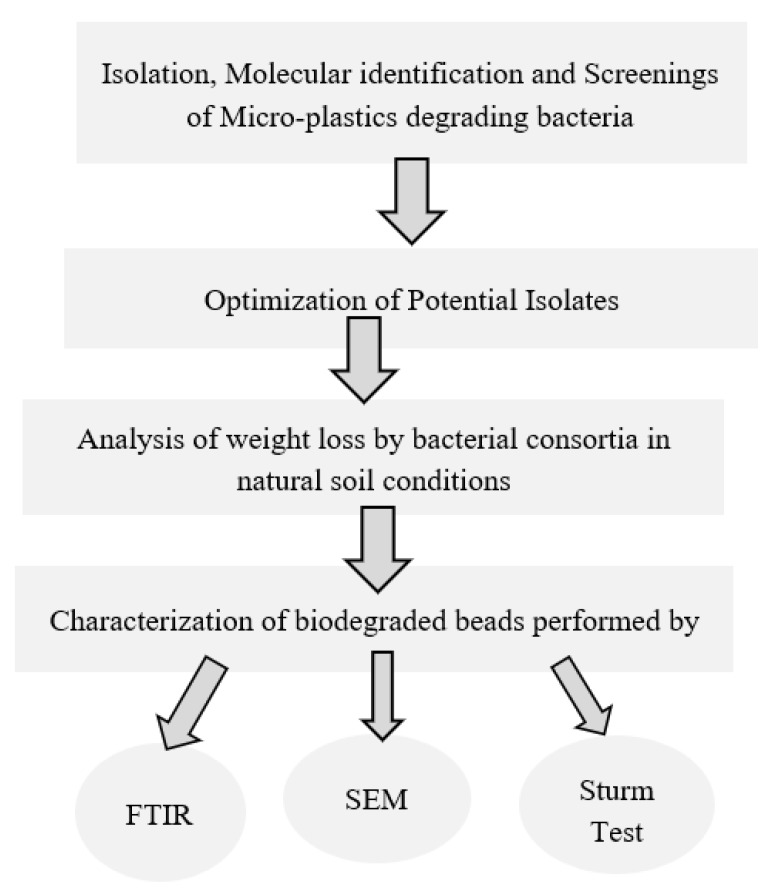
Schematic diagram of the study.

**Figure 2 polymers-14-02275-f002:**
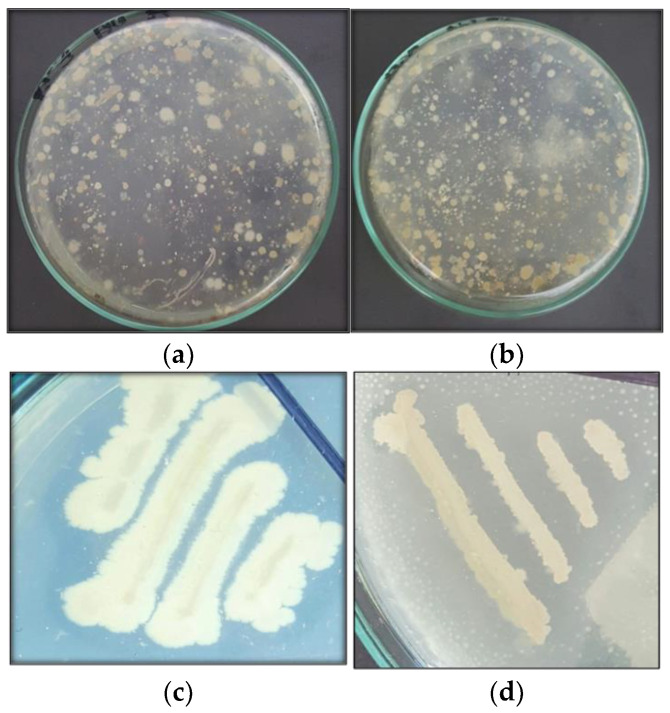
(**a**,**b**) showing growth of bacterial colonies on LLDPE and HDPE with clear zones, (**c**,**d**) showing growth of bacterial isolates on LLDPE and Polyester.

**Figure 3 polymers-14-02275-f003:**
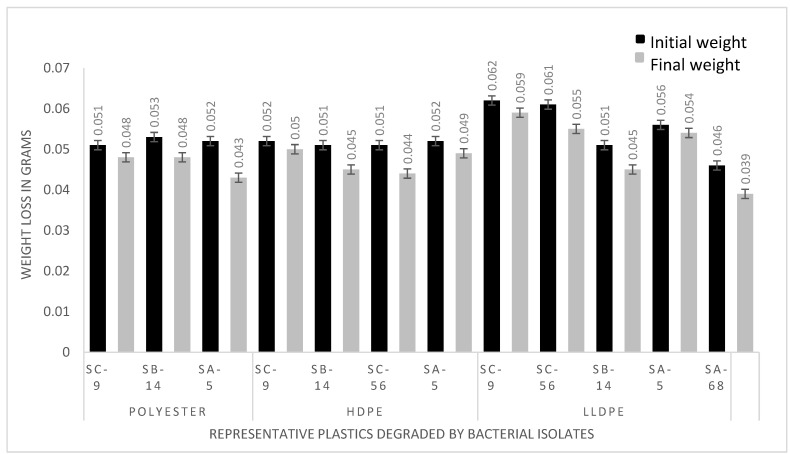
Significant weight reduction in plastic by bacteria—results shown by potential isolates.

**Figure 4 polymers-14-02275-f004:**
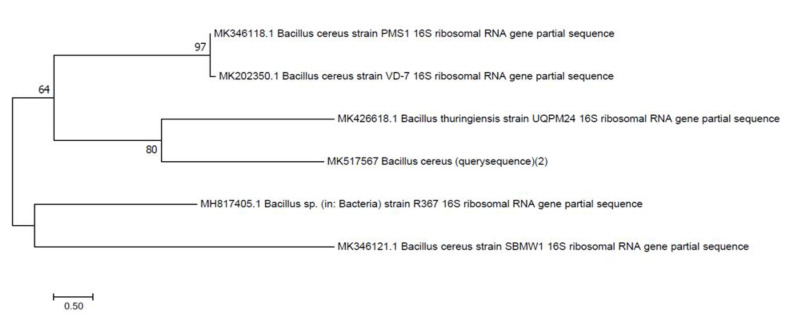
Neighbor joining tree of *Bacillus cereus* (MK517567).

**Figure 5 polymers-14-02275-f005:**
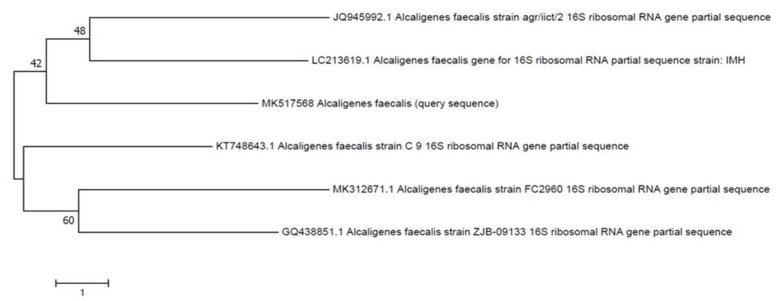
Neighbor joining tree of *Alcaligenes faecalis* (MK517568).

**Figure 6 polymers-14-02275-f006:**
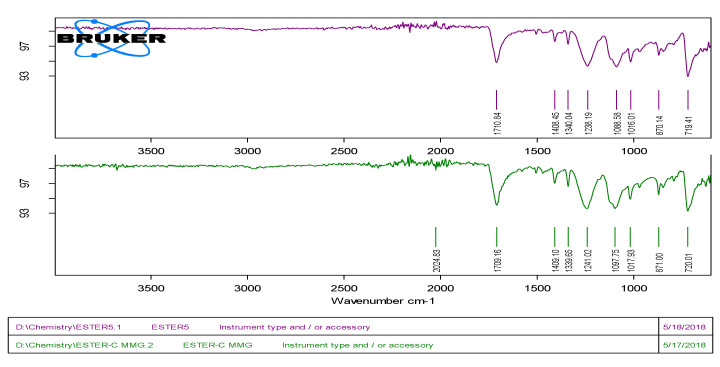
Spectral peaks of Polyester indicating the comparison of *Alcaligenes faecalis* (SA-5) with control.

**Figure 7 polymers-14-02275-f007:**
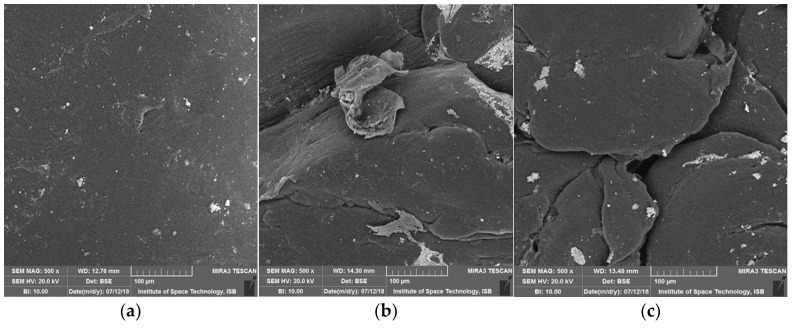
SEM captures of LLDPE beads following 40 days of incubation by isolates. (**a**) Control, (**b**) SA-5 Alcaligenes faecalis treated LLDPE showing bacterial attachment, (**c**) SA-68 Bacillus cereus treated LLDPE showing surface cracks, (**d**) SB-14 treated LLDPE showing surface erosion, (**e**) SC-56 treated LLDPE showing granules of bacterial action, (**f**) SC-9 treated LLDPE showing pith and groove.

**Figure 8 polymers-14-02275-f008:**
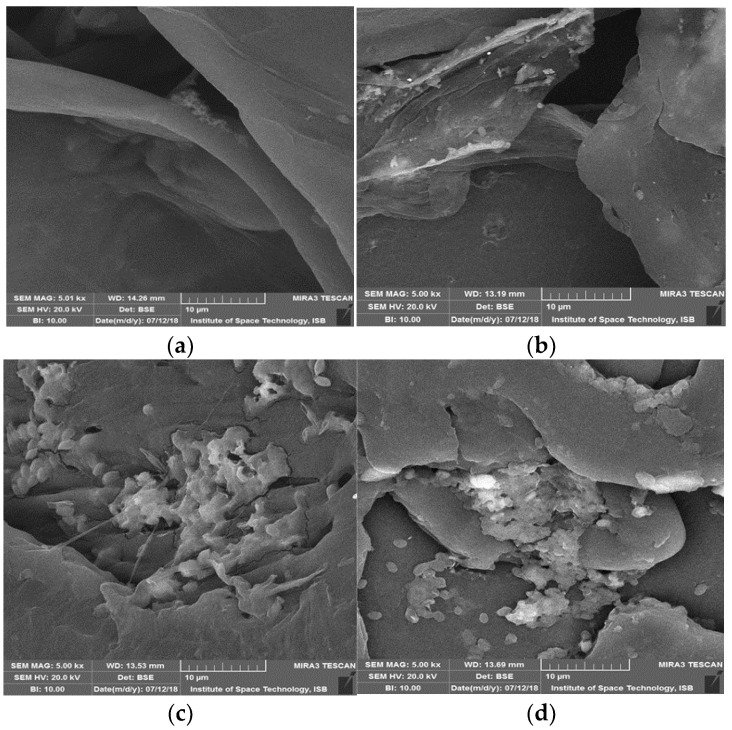
SEM captures of HDPE beads following 40 days of incubation by isolates. (**a**) Control showing no bacteria present, (**b**) SA-5 Alcaligenes treated HDPE showing groove, (**c**) SB-14 treated HDPE showing bacterial granules, (**d**) SC-56 treated HDPE showing surface cracks and bacterial attachment.

**Figure 9 polymers-14-02275-f009:**
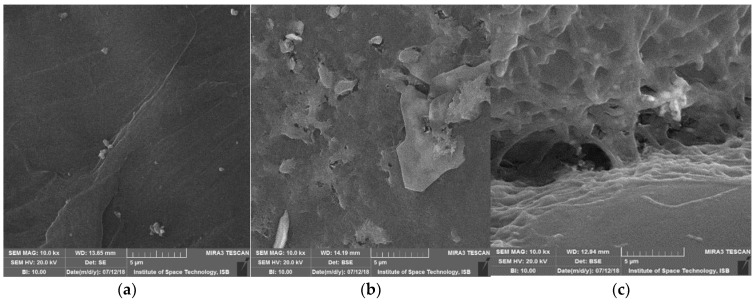
SEM captures of polyester beads following 40 days of incubation by isolates. (**a**) control-smooth surface, (**b**) SA-5 Alcaligenes treated polyester showing piths, (**c**) SB-14 treated polyester showing groove, piths and holes.

**Table 1 polymers-14-02275-t001:** FTIR peaks corresponding to vibrational moods and functional groups of LLDPE and HDPE.

**Spectral Peaks Shift in LLDPE**				
Peak Number	1	2	3	4
Frequency (cm^−1^)	2913.83	2846.50	1462.05	718.23
Vibrational mood & Functional Group	C–H stretching of methyl group	C–H symmetric and asymmetric stretching of methylene (C–H2)	C=C and replacement of carbonyl bond with amine bond	–C=C– stretching and the presence of alkene group
**Spectral Peaks Shift in HDPE**				
Peak Number	1	2	3	4
Frequency (cm^−1^)	2914.37	2846.70	1461.76	717.82
Vibrational mood & Functional Group	stretching of C=C bond	stretching of C-H bonds in methylene	–CH2 stretching	C=C– stretching

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
