# Peer review of "Biodeterioration of Microplastics: A Promising Step towards Plastics Waste Management"

_polymers, 2022, doi:10.3390/polym14112275_

Round 1

Reviewer 1 Report

The manuscript Biodeterioration of Microplastics: a Promising Step Towards Plastics Waste Management, by Aatikah Tareen, Saira Saeed, Atia Iqbal, Rida Batool and Nazia Jamil, submitted for publication in the Journal Polymers, section: Biomacromolecules, Biobased and Biodegradable Polymers, special issue Advances in Biocompatible and Biodegradable Polymers, presents an original study on biodegradation of some microplastics – Linear Low Density Polyethylene, High Density Polyethylene and Polyester using local bacterial colonies of Alcaligenes faecalis (MK517568), Bacillus cereus (MK517567) and Alcaligenes faecalis (MK517568). Soil bacteria of these strains are successfully isolated, characterized and applied in adequate experiments for plastic degradation, including soil burial method. The effect of polymer deterioration is proved also by means of FTIR spectroscopy, Sturm test and Weight loss analysis and visualized by Scanning Electron Microscopy. The potential of the examined types of bacterial communities in the plastic waste management is demonstrated.

The results reported are of substantial fundamental and applied interest as they are closely related to the ability of plastics waste control, by its destruction, neutralization and utilization, thus tangenting to the ecology and circular economy.

However, I have the following remarks and comments:

  1. The main strategies for plastics waste removal in general are discussed in the Introduction The recycling is one possible way for this removal and it is widely applied, together with burning and reuse. The authors have to include also some comments on recycling of polymers in the Introduction.
  2. The microplastics, being an object of the current investigation, are involved in the Introduction, by the definition sentence: “These scrapes are characterized as microplastics (0.1 μm-5 mm) based on their diverse range.”, which is not correctly expressed because the essence of the notion is not revealed. As “microplastics” are defined polymer (or plastic) particles, which sizes are in the range: 1 μm-5 mm.
  3. Optical images of the bacterial colonies should be added.
  4. In the subsection “Formulation of microbial consortia and determination of weight loss for mixture of plastic pellets in local natural conditions” a final sentence/sentences for the weight loss detection and determination should be added for completeness.
  5. I recommend changing the locations of the results for SEM and FTIR analyses in the section Results. As the SEM visualized the results for plastics biodegradation and as the authors mention them self that SEM confirms them, the figures with SEM micrographs could be logically at the end of the section.

Many typos and inexactitudes are also detected and the authors must check and correct the text very carefully. For example:

  1. Lack of intervals in many sentences.
  2. Missing essence of some sentences. Ex.: page 2 “The strategies given, by screening with electron microscopy and Fourier-transform infrared spectroscopy.”
  3. page 4: “An acclimated consortium capable of degrading polyester, high-density and linear low-density polyethylene pellets developed in laboratory.” – a missing verb
  4. page 4: “To examine biodegra-dation of LLDPE, HDPE and Polyester in ex-situ condition.” – must be edited
  5. Sequence of tenses is needed.
  6. A text and symbols or box borders overlapping is detected in Figure 1.
  7. The format of Figure 8 is not appropriate.

Author Response

Reviewer Comments, Author Responses and Manuscript Changes

Reviewer # 1

Comment 1:The main strategies for plastics waste removal in general are discussed in the Introduction. The recycling is one possible way for this removal and it is widely applied, together withburning and reuse. The authors have to include also some comments on recycling ofpolymers in the Introduction.

Response: Author added that “Currently limited quantities are reused and this causes prevention to the idea of circular economy, as only 9% of plastic was recycled from the total production of 8.3bn metric tons since 1950’s. It was also investigated that in 1990, Coca-Cola promised to use 25% of recycled plastic by 2015, but in 2020 only 11.5% had been recycled.

Comment 2:The microplastics, being an object of the current investigation, are involved in theIntroduction, by the definition sentence: “These scrapes are characterized as microplastics(0.1 μm-5 mm) based on their diverse range.”, which is not correctly expressed because theessence of the notion is not revealed. As “microplastics” are defined polymer (or plastic)particles, which sizes are in the range: 1 μm-5 mm.

Response: Correction applied.

Comment 3: Optical images of the bacterial colonies should be added.

Response: Optical images for bacterial colonies with zones and the purified bacterial isolates growth on plastics supplemented media was added to manuscript as a figure 2.

Comment 4:In the subsection “Formulation of microbial consortia and determination of weight loss formixture of plastic pellets in local natural conditions” a final sentence/sentences for the weightloss detection and determination should be added for completeness.

Response: We added up the sentence and every time we took pre-weighted samples and after the experiments, we did weight of that sample then put the initial weight and final weight in formula (mentioned in manuscript) to find weight loss percent.

Comment 5:I recommend changing the locations of the results for SEM and FTIR analyses in the section. Results. As the SEM visualized the results for plastics biodegradation and as the authorsmention them self that SEM confirms them, the figures with SEM micrographs could belogically at the end of the section.

Response: We appreciate your recommendations and it was applied.

Many typos and inexactitudes are also detected and the authors must check and correct the textvery carefully. For example:

1.Lack of intervals in many sentences.

Corrected

2.Missing essence of some sentences. Ex.: page 2 “The strategies given, by screening withelectron microscopy and Fourier-transform infrared spectroscopy.”

Corrected

3.page 4: “An acclimated consortium capable of degrading polyester, high-density and linearlow-density polyethylene pellets developed in laboratory.” – a missing verb

Corrected

4.page 4: “To examine biodegra-dation of LLDPE, HDPE and Polyester in ex-situ condition.” –must be edited

Corrected

5.Sequence of tenses is needed.

6.A text and symbols or box borders overlapping is detected in Figure 1.

Corrected

7.The format of Figure 8 is not appropriate.

Corrected

Reviewer Comments, Author Responses and Manuscript Changes

Reviewer 2 Report

This article intends to address the important issue of plastic degradation with a focus on studying plastic biodegradation in soil environment. This article may be acceptable for Polymers given if the following issues can be addressed:

  1. Please clearly label the references numerically in the manuscript to match the references given at the end of the article. The current form of referencing is difficult for readers to track.
  2. For sample selection, does "aromatic polyester" refer to PET? Please be more specific to explain the structure of the experimental samples to readers.
  3. Major explanation/revision needed from the authors: The reviewer is not entirely convinced by the FTIR analysis, which is a critical part of this article. Most the shifts in peaks are very small: for example, 2913.92 (control) shifts to 2913.83cm-1. Some of these shifts might be even within the instrumental errors. The authors should address this. Have these observations previously been reported in similar literatures on PE or PET deconstruction that also show small shifts in IR peaks?
  4. For polyester degradation, the authors claimed the C-O bond cleavage by IR analysis. If this has occurred, should the authors observe the formation of O-H stretches in the product IR? The formation of either -COOH or -OH should be expected from an ester cleavage. Please explain.
  5. Please give a conclusion section to summarize your work.

Author Response

Reviewer Comments, Author Responses and Manuscript Changes

 Reviewer # 2

Comment 1:Please clearly label the references numerically in the manuscript to match the referencesgiven at the end of the article. The current form of referencing is difficult for readers to track.

Response: Although changes were done according to your suggestion, references labelled numerically.

Comment 2:For sample selection, does "aromatic polyester" refer to PET? Please be more specific toexplain the structure of the experimental samples to readers.

Response:Thanks for your concern, structure mentioned in materials. Here we used the pure form of polyester, the form of polyester used as fibers.

Comment 3:Major explanation/revision needed from the authors: The reviewer is not entirely convincedby the FTIR analysis, which is a critical part of this article. Most the shifts in peaks are verysmall: for example, 2913.92 (control) shifts to 2913.83cm-1. Some of these shifts might beeven within the instrumental errors. The authors should address this. Have theseobservations previously been reported in similar literatures on PE or PET deconstruction thatalso show small shifts in IR peaks?

Response:In addition to the above comments, authors tried to add relevant and latest reference in discussion section and literature also identified the same peaks or vibrational ranges as we mentioned them. In some cases the vibrational shifts are very small because we did this experiment for only 40 days, that’s why some bacterial isolates gives low percentages, otherwise in case of polyester degradation by Alcaligenes faecalis (SA-5) show the weight loss by 17% and in FTIR spectra, the removal of peak also takes place.

Comment 4: For polyester degradation, the authors claimed the C-O bond cleavage by IR analysis. If thishas occurred, should the authors observe the formation of O-H stretches in the product IR?The formation of either -COOH or -OH should be expected from an ester cleavage. Please explain.

Response: According to your suggestions, authors try to explain it in discussion section by adding reference.

Comment 5: Please give a conclusion section to summarize your work.

Response: Thanks for concern, conclusion is added.

Round 2

Reviewer 1 Report

The text of the manuscript has been improved according to the reviewers' recommendations. There are still typos and need for editing. 

Reviewer 2 Report

This revised manuscript in its revised form can be accepted by Polymers.